# Temperature and feeding frequency impact the survival, growth, and metamorphosis success of *Solea solea* larvae

Adriana E. Sardi[1]*, Marie-Laure Bégout[2], Anne-Laure Lalles[1], Xavier Cousin[2], Hélène Budzinski[1]

1 CNRS, EPOC, UMR 5805, University of Bordeaux, Talence, France, 2 MARBEC, CNRS, Ifremer, IRD, INRAE, Univ Montpellier, Palavas-Les-Flots, France

* adrianasardi@gmail.com

**Editor:** Sergio N. Stampar, Sao Paulo State University Julio de Mesquita Filho Bauru Campus Faculty of Sciences: Universidade Estadual Paulista Julio de Mesquita Filho Faculdade de Ciencias Campus de Bauru, BRAZIL

## Abstract

Human-induced climate change impacts the oceans, increasing their temperature, changing their circulation and chemical properties, and affecting marine ecosystems. Like most marine species, sole has a biphasic life cycle, where one planktonic larval stage and juvenile/adult stages occur in a different ecological niche. The year-class strength, usually quantified by the end of the larvae stage, is crucial for explaining the species' recruitment. We implemented an experimental system for rearing larvae under laboratory conditions and experimentally investigated the effects of temperature and feeding frequencies on survival, development (growth), and metamorphosis success of *S. solea* larvae. Specific questions addressed in this work include: what are the effects of feeding regimes on larvae development? How does temperature impact larvae development? Our results highlight that survival depends on the first feeding, that the onset of metamorphosis varies according to rearing temperature and that poorly fed larvae take significantly longer to start (if they do) metamorphosing. Moreover, larvae reared at the higher temperature (a +4°C scenario) showed a higher incidence in metamorphosis defects. We discuss the implications of our results in an ecological context, notably in terms of recruitment and settlement. Understanding the processes that regulate the abundance of wild populations is of primary importance, especially if these populations are living resources exploited by humans.

## Introduction

The Intergovernmental Panel on Climate Change (IPCC) has been, since its first report back in 1990, actively outlining the current and future effects of climate change (CC) and the actions needed for preventing reaching the point of no return. At the current emission rate, the 1.5°C threshold will be exceeded by 2030 to 2052, and a 3–4°C temperature increase is predicted by 2100 [1]. Further, future projections performed within the Intergovernmental Panel on Climate Change (IPCC) context indicate a significant global reduction of primary production with critical consequences on fisheries and marine biodiversity [2, 3]. In their report from

**Data Availability Statement:** All relevant data are within the paper and its Supporting information files.

**Funding:** A.S. was funded by the IdEx Bordeaux International Post-doctorates program. The funders had no role in study design, data collection and analysis, decision to publish, or preparation of the manuscript.

**Competing interests:** The authors have declared that no competing interests exist.

2018, the authors underline that the consequences of global warming of 1°C are genuine, particularly the increased occurrence of extreme weather events, the rise in sea level, and the decrease in Arctic sea ice [4]. Following the release of the most recent report, leading author, Dr. Joeri Rogelj, highlights that this report is likely to be the last one while there is still time to stay below the 1.5°C threshold (The Guardian, Major climate changes inevitable and irreversible-IPCC's starkest warning yet- published 9/08/2021).

Most marine species, including teleost fishes, have a multiphasic life cycle, where one -or multiple- planktonic larval stages and juvenile/adult stages occur in a different ecological niche [5]. The complexity of life cycles with multiple and distinct phases is thought to promote higher dispersion of individuals due to oceanic currents followed by reduced predation and justified by access to a larger food source [5]. However, having different niche on different life stages also means that different stages will be exposed to different scenarios, making potentially harder the transition from one stage to another.

Compared to any other life stage of a marine fish species, individuals at their larval stage will have the highest potential for growth, weight-specific metabolic rates, natural mortality rates, and the highest sensitivity to environmental stressors [6, 7]. Early stages have strict environmental requirements, where nutrition, microbial environment, and physical/chemical conditions determine the healthy development and the survival rate. Therefore, when these environmental requirements are not met, there are consequences at later life stages that could compromise the fitness of individuals and thus the health of populations.

Larvae survival has been hypothesized to be fundamental for fish recruitment variability. In 1914 Johan Hjort proposed the "critical period hypothesis", which orbits around the idea that the fate of fish year classes is determined during the early larval stage, specifically shortly after yolk absorption when fish larvae must have found suitable prey in order to survive. The match-mismatch hypothesis [8], which extends from Hjort's hypothesis, proposes that fish spawning times and survival may be related to the zooplankton production cycle [8]. Under the match-mismatch hypothesis changes induced by temperature rise at the time of spawning would impact the interactions between dispersal patterns and the spatial and temporal dynamics of plankton biomass, which are critical for larvae survival. Unfortunately, evidence suggests that the time for spawning is being advanced by temperature [9] and this shift in sole spawning phenology might alter the matching between ichthyoplankton and their food, particularly in areas where algal blooms are triggered by wind regimes and light as in the North Atlantic Ocean [10].

The flatfish species *Solea solea* (Fig 1), is an important benthic commercial species found in shallow (10–60 m and up to 200 m depth) estuarine and coastal waters of the eastern Atlantic Ocean [11]. Sole has a complex life cycle, and environmental fluctuations strongly influence the time of spawning, the survival rates, egg and larval life history along the larval dispersal process, and the transition from a pelagic to a benthic lifestyle [12]. Further, sole shows progressive habitat shifts along ontogeny. Shortly, fertilized eggs and larvae are both pelagic and occur in the continental shelf, which allows for wide dispersion and exposes them to environmental variability, notably temperature and food abundance. At 16°C, fertilized eggs hatch after five days [13], and while recently hatched larvae drift to estuaries and coastal areas -their juvenile environment- they undergo metamorphosis. Significant ontogenetic, morphological, physiological, and behavioral changes occur during metamorphosis [14]. The length of metamorphosis varies depending on temperature and food availability. The transition from hatching until complete metamorphosis -the complete larval stage- can last between 26 and 15 days, at 16°C and 19°C, respectively [11]. By the end of metamorphosis, juvenile soles are ready to change from pelagic to benthic lifestyle to colonize shallow coastal waters in estuaries and bays [15]. After settlement, juveniles stay for around two years until they reach maturity. Adults

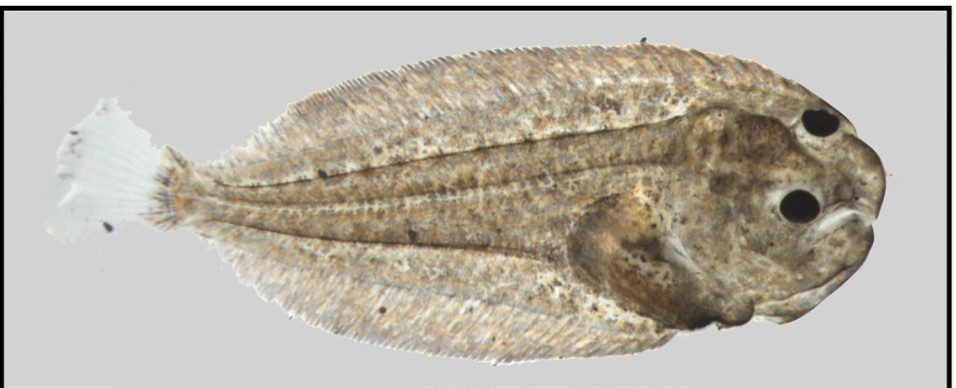

**Fig 1. Fully metamorphosed *Solea solea* larva.** A fully metamorphosed larva showing full asymmetrical body, rounded head and complete left eye migration.

migrate offshore (few or up to hundreds of kilometers depending on location) for reaching spawning grounds, and they eventually come back closer to the shore out of reproduction season to reach feeding grounds [11]. To summarize, three distinct life stages characterize the development of soles: a post-hatch pelagic larval phase, a juvenile benthic phase, and an adult benthic phase. Such a complex life cycle makes it challenging to determine how habitat quality affects population dynamics.

As an r strategist species and a batch spawner, *S. solea* is characterized by a high fecundity. The spawning of hundreds of thousands of eggs per kilo of female with overall low survival rates is triggered by changes in water temperature trigger [13].

The reproductive success of soles -quantified as the number of eggs, larvae, or juveniles that survive and settle in coastal nursery areas- widely depends on hydrodynamic processes, and it is regulated by environmental factors, among them, temperature [16]. While ecological processes acting in nursery areas can affect year-class survival, these are generally considered less important than those influencing larval stages. This assumption relies on the fact that nurseries are stable habitats that provide suitable conditions for the survival of juveniles [16].

Understanding the processes that regulate fish recruitment are of primary importance to assess the abundance of wild populations, regulate fisheries, and prevent alterations in the ecosystem functioning. Two of these factors -often cited as underlying growth and survival- are competition for limited resources (food) and predation. In the case of juvenile sole, research suggests that competition for limited resources, particularly food, seems to be one of the most likely reason explaining survival rates in coastal nurseries rather than predation pressure on larval and juvenile fish [17]. However, less information is available for the impacts of varying food levels on larvae development and recruitment variability, and only few and recent studies have included restricted feeding regimes into their experimental designs (see [18]).

We now generally acknowledge that no single process, mechanism, or factor is responsible for recruitment variability but that many processes may act together over the entire egg to the pre-recruit juvenile period [19]. Among the multiple processes involved, dominant recruitment mechanisms include temperature, prey, predation, physical processes, and the bigger and faster relationship [19]. The last is the foundation of the "stage duration" hypothesis, which implies that large size and fast development is better as it reduces predation pressure and improves survival [19].

We hypothesize that low food availability will delay the growth and metamorphosis of sole larvae. At the same time, the temperature increase will fasten up the ontogenesis provided that

there is enough food to compensate for the rise in energy demands a higher metabolism implies. In the wild, reduced food availability or starvation would slow metamorphosis and increase predation risk, thus reducing larval survival.

In this work we experimentally investigated the combined effects of temperature and feeding frequencies on the survival, development (growth), and metamorphosis success of *S. solea* larvae. Our objective was to disentangle the effects of food availability and increased temperatures -imitating future CC predictions- on larvae development. Specific questions addressed in this work include i) what are the effects of different levels of food availability -approximated by different feeding frequencies- in larvae growth and metamorphosis? ii) how does temperature interact with food levels and impact larvae growth and metamorphosis? And based on these results iii) what consequences of metamorphic defects on *S. solea*'s recruitment can be foreseen?

## Materials and methods

In all our experiments, we used 24-well polystyrene microplates with a cover as a housing system. This rearing system is a simplified version -that does not include gnotobiotic conditions- of the method developed by De Swaef and collaborators [20]. Advantages of using microplates for rearing fish larvae include the possibility of monitoring sole larvae growth, development, and survival at the individual level during the whole experiment. In addition, using microplates assures that the death of a larva will not affect the status of the other ones. The Research and High Education French Ministry evaluated and approved all animal experimentations. The committee from the department of practical research activities licensed the project "Flatfish adaptation to temperature, feeding and pollution stress" led by Dr. Marie-Laure Bégout from Ifremer under the number reference APAFIS #38190–202208082103879.

### Experimental set-up to house common sole larvae in 24-well plates

To ensure the viability of using microplates as a housing system, we ran a pilot experiment in April 2019. The objectives were to test the rearing system -particularly the survival of larvae- and to determine larvae' maximum food intake per day. We obtained *Solea solea* eggs from a broodstock in the Netherlands (Zeeschelp, Kamperland). Sole adults naturally spawned in several batches from the 22nd to the 24th of April 2019 at the facilities in the Netherlands. Eggs arrived at the Ifremer facilities in La Rochelle, France, on the 25th of April. Once in the lab, fertilized eggs were sorted under the stereomicroscope (Olympus, SZX9) and transferred to glass bottles containing clean and filtered (GF-C 0.45μm, Millipore) natural seawater at a salinity of 30.

The bottles with fertilized eggs were placed in an environmental chamber (Sanyo MLR-351) with constant temperature (16 ± 1˚C), oxygen and photoperiod conditions (12h light, 12h darkness). Hatching occurred during the weekend of 27th and 28th of April (between 4- and 5-days post-fertilization, depending on the batch).

Two days post-hatching (dph), 72 larvae were transferred to 24-well polyethylene microplates (Sigma-Aldrich, CLS3527-100EA). Before transferring, we prepared the plates by adding 1950 μl of the water media. Larvae were gently pipetted on a volume of 50 μl and transferred to the wells. We used a 200 μL capacity micropipette with a tip cut to transfer the newly hatched larvae and avoid damage to the entering larvae.

All microplates were placed within an environmental chamber and incubated on the same conditions described above. Survival, malformations, behavior, and metamorphic stages, were monitored daily using an Olympus SZX9 stereomicroscope. The experiment lasted for 33

**Table 1. Main differences between the pilots and the experiment conducted.**

| | Pilot 1 | Pilot 2 | Experiment |
|---|---|---|---|
| Objectives | Testing survival in the microplates as housing system. Determining the amount of artemia for feeding *ad libitum* | Determining the timing for starting the temperature treatments. Setting the different food treatments. | Determining the combined effects of temperature and feeding frequencies in sole larvae |
| Larvae exposed at 20˚C (age) | | Immediately after placing larvae in microplates (4–5 dph) | Temperature increased gradually, 1 degree per day from day 8 post-hatching (larvae at 20˚C from 11 dph onwards) |
| Food availability at mouth opening day | *Ad libitum.* The quantity of food provided and consumed per larvae each day was monitored (counted) | Different food densities among treatments HighFood = 5 artemias MediumFood = 3 artemias LowFood = 2 artemias | Same food density and feeding frequency during the first 7 days following mouth opening (10 artemias per larvae). From 12 dph onwards, we changed the feeding frequency among treatments (with an increasing number of artemias 15 to 40) the following way: HighFood: fed every day MediumFood: fed every other day LowFood: fed twice a week |

days. Water was changed every other day by carefully removing as much as possible of the water (~1800 μl) while moving the tip away from the larvae to avoid suctioning it.

From day 5 post-hatching onwards, we fed larvae with *Artemia salina* (Sep-Art Artemia Cyst, Ocean Nutrition). We determined the number of artemias eaten per larvae by monitoring the quantity of food provided and consumed per larvae each day. On the first day, larvae were fed five artemias. The amount of artemia provided increased following the percentage of daily food consumed. The interest was providing food *ad libitum* without compromising the water quality. Please refer to S1 Table in S1 File to know the exact amount of artemias given to the larvae daily until the end of the experiment.

## Experimental set-up to test the effects of temperature and feeding regimes

In 2021, we obtained fertilized eggs from the same broodstock, and to determine the timing for starting the temperature treatments, we ran a second pilot experiment, which was immediately followed by the experiment. Eggs employed in the second pilot (Table 1) spawned from the 26th to the 28th of April, and eggs arrived at the Ifremer Palavas-les-Flots, France the 29th of April 2021. Eggs employed in the experiment spawned the 8th of May, and arrived at Ifremer the 12th of May. In both occasions, fertilized eggs were sorted and transferred to glass bottles containing freshly prepared artificial seawater at a salinity of 30 and a temperature of 16˚C. Artificial seawater was prepared by diluting 30 g of salt (Instant Ocean) in type II water, and we adjusted salinity to 30 using a salinometer. Further, we filtered the water using GF-C 0.45 μm to remove non-dissolved particles.

Two days post-hatching (dph), 144 larvae were transferred to six 24-well polyethylene microplates. For the pilot experiment we equally distributed larvae from different batches on the experimental microplates while in the experiment all larvae were from the same spawning event (8th of May). Procedures for transferring larvae to microplates and monitoring survival and development were the same as in the first pilot experiment. The experiment lasted until larvae were 35 dph age; which roughly corresponds to the age in the experiments ran by De Swaef et al. in 2017 (experiment ended at 26 dph).

## Experimental design

The experimental design for pilot 2 and the experiment included two experimental factors, temperature with two levels, 16˚C and 20˚C, and a feeding (density in the pilot and frequency

in the experiment) factor, which was assumed to be a proxy for food availability (Fig 2a). The last factor included three levels: high food, medium food, and low food, from now on indicated in the text as HighFood, MediumFood and LowFood, respectively. The main differences between the two pilots and the experiment are summarized in Table 1.

In each experiment, two groups of microplates were designed. The first group, called the experimental group, consisted of 3 microplates and 72 larvae. Each temperature treatment included three feeding regime levels. Larvae from the experimental group were planned for monitoring survival and metamorphosis daily. The second group, called the biometric group, included 3 additional microplates and 72 larvae, with the same treatments as the experimental group. These larvae were planned for monitoring changes in length and dry weight along the experiment. For that, a total of four larvae per plate were sacrificed every week. We euthanized the larvae using a 100 g L$^{-1}$ benzocaine solution at a lethal concentration of 500 mg L$^{-1}$.

## Effects of temperature and food availability

To study the effects of temperature on sole larvae growth and development, we incubated larvae at 16˚C, an optimal temperature for *S. solea* larvae development, and at 20˚C, a + 4˚C condition imitating the IPCC 8.5 scenario [21]. The optimal temperature of 16˚C was chosen based on the methods employed in previous aquaculture-oriented experiments rearing *S. solea*, notably the work from De Swaef et al. 2017.

In pilot 2 larvae were directly placed in the environmental chamber at 20˚C following hatching and the food availability treatments consisted in different food densities. This pilot experiment allowed us to improve the method of larvae acclimation to the 20˚C temperature treatment. In the experiment larvae were acclimated with a gradual temperature increase (Fig 2b). For this, we increased the temperature of the environmental chamber by one degree every day from day 8 post-hatching, which corresponded to three days after the first meal or mouth opening day.

The effects of food availability were approximated and tested by comparing larvae metamorphosis, survival and growth from larvae reared with different feeding frequencies. To feed recently hatched larvae we provided A0 *Artemia salina* (less than 24h old individuals). Because recently hatched larvae are too small to eat enriched A1 artemias, we decided to provide unenriched food along the whole experiment.

In the second pilot, the quantity of *A. salina* provided from mouth opening day differed between treatments, with HighFood treatment receiving an amount of artemia assumed to be *ad libitum*. The amount of artemia *ad libitum* per age was obtained from our first pilot experiment. We calculated the amount of artemias for the other two treatments representing 60% and 40% of the HighFood treatment. Further, the number of artemias provided each day gradually increased to accommodate the energy needs of growing individuals (S2 Table in S1 File).

During the experiment, we fed equally all larvae every day during the first five days post mouth opening. From 12 dph onwards, we changed the feeding frequency among treatments as it was a simpler approach than altering the food density. HighFood larvae were fed 6 days a week, while MediumFood and LowFood treatments were fed three times and twice a week, respectively. On 12 dph, we provided around 15 artemias, and as in pilot 2 the amount of food gradually increased along the experiment.

## Metamorphosis

The evolution of metamorphosis was monitored six times a week during the whole experiment. The metamorphosis stages were evaluated using the following criteria [22]:

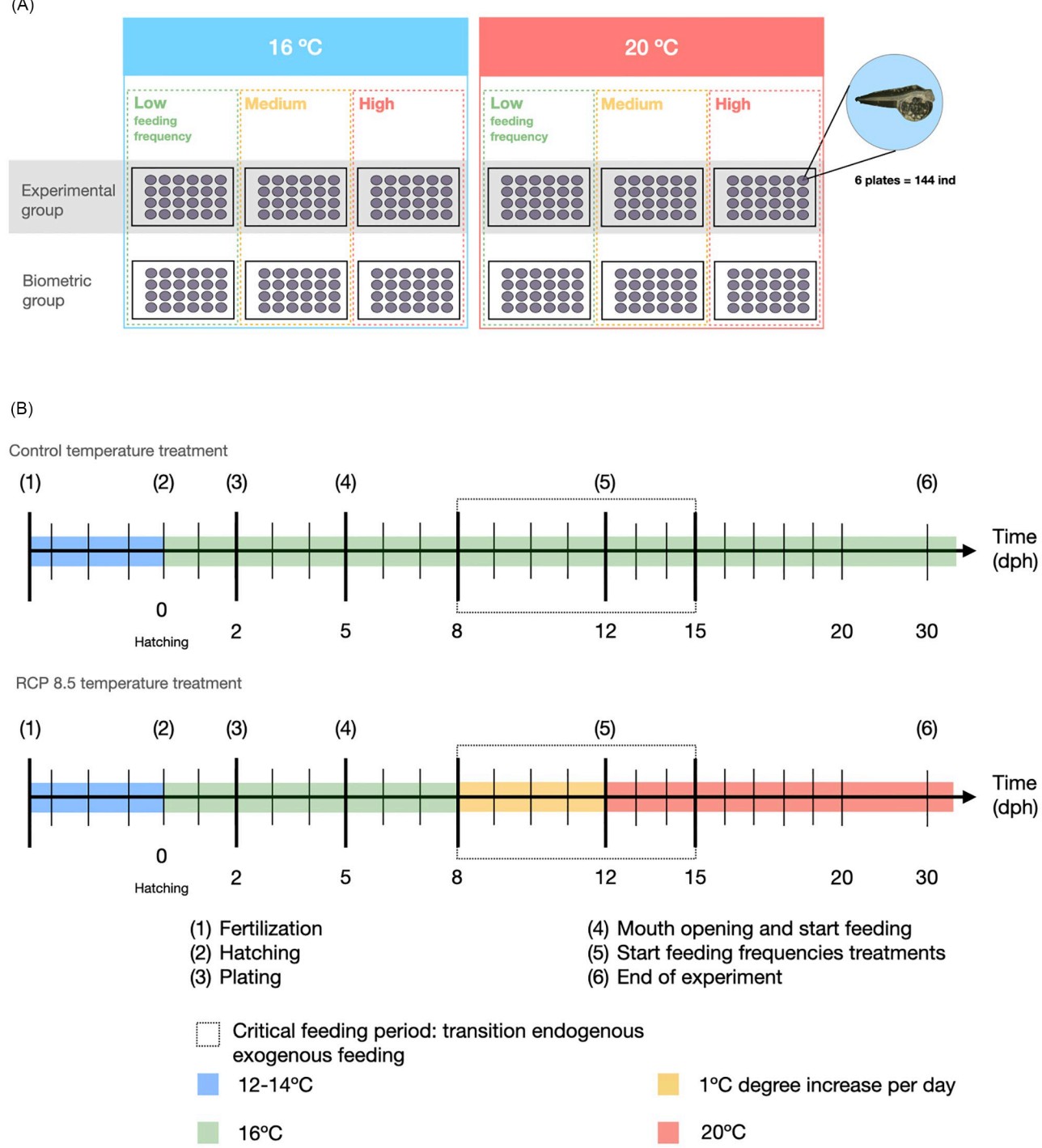

**Fig 2. Experimental design.** Number of plates used per treatment at both experiment 1 and 2 (a) and timeline protocol followed during experiment 2, for larvae reared at 16˚C (control temperature treatment) and at 20˚C (RCP 8.5 temperature treatment), that included a temperature acclimation for larvae.

- stage 0: before the start of metamorphosis, when the larva is symmetrical to a vertical swimming plane

- stage 1: the left eye separates from the eye socket and begins to migrate to the dorsal position

- stage 2: when the left eye is close to the midline of the dorsal face; at this stage, the individuals are not completely symmetrical

- stage 3: the left eye goes beyond the dorsal midline and can be observed from the top side; at this stage, all individuals move to the bottom

- stage 4: the eye migration is finished, the head and the dorsal midline are completely oval and reshaped

## Length and weight measurements

The length and dry weight of a total of 4 larvae per treatment were measured at 8, 15 and 22 dph. These larvae were sacrificed and collected from the biometric group microplates described before. We euthanized the larvae and placed them one by one on a microscope calibration slide (grid 1 mm). Because larvae survival was lower than anticipated, from week 4 until week 7 of the experiment, only length measurements without killing the larvae were taken. We did so by simply reducing the water volume at each well and taking a picture for each individual and a picture of a calibration slide. Pictures to quantify length were taken using the same stereomicroscope connected to a camera (The Imaging Source, DFK 31AU03). For all length pictures, color lighting was set to 3000 K (Schott KL 1500 LCD cold light source), and the same stereomicroscope magnification. All length measurements were performed with the free Java image processing program ImageJ [23].

## Data processing

Data were input into the R environment [24]. To do the figures, we used packages *ggplot2* [25] and *ggpubr* [26].

Differences in mean size and dry weight of larvae reared under different temperature and feeding conditions were evaluated using Euclidean distances as a measure of dissimilarity and tested through PERMANOVA [27, 28]. The fundamentals of the PERMANOVA test are similar to ANOVA's; however, it uses permutations to calculate p-values, rather than relying on tabulated p-values that assume normality of the data [28]. The only true assumption in PERMANOVA is that data must be independent and not correlated with each other (temporally or spatially). This last point justifies our decision of excluding the notion of time (*i.e.*, comparing larvae size for different ages).

The testing design consisted of 2 factors orthogonal to each other, feeding density in experiment 1 or feeding frequency in experiment 2 (fixed, three levels, HighFood, MediumFood, and LowFood), temperature (fixed, two levels, 16˚C, and 20˚C), and their interaction. If the interaction was not significant, differences among feeding frequencies were retested for each temperature separately. For the PERMANOVA tests we used the package *vegan* [29] and significant terms in the model ($\alpha = 0.05$) had their means compared with a PERMANOVA pairwise *a posteriori* test [30] using the *Bonferroni* p-value adjustment method.

For comparing survival, we used the package *survival* [31, 32] to do the Mantel-Cox test, and the results were represented using Kaplan-Meier curves done with the package *survminer* [33]. The Mantel-Cox test or log-rank test is a non-parametric and hypothesis-based test that allows comparing the distribution of survival curves of at least two samples. The test

is convenient when the data are asymmetric or censored (in our case, the censored data correspond to the individuals taken for biometrics). Finally, to visually compare the survival patterns, we fitted a linear model to the obtained survival from experimental days 8 to 15, where we observed the more drastic mortality.

We determined the $LT_{50}$ for each treatment, defined as the time for 50% of the individuals to die. For that, survival data of each treatment combination was fitted to a four parameters logistic curve, using Prism (Graphpad).

## Results

### Survival

During the first days of the experiment, the number of surviving larvae decreased considerably, being the larvae reared at 20˚C LowFood, the ones that died faster. Both temperature treatments had the same decreasing tendency, and the lethal time for 50% of the individuals ($LT_{50}$) happened in average 2 days earlier in larvae reared at 20˚C vs. larvae reared at 16˚C (Table 2). At 20˚C, the lowest $LT_{50}$ was obtained for the MediumFood treatment, which was significantly different from Low and HighFood. No differences were detected between the $LT_{50}$ obtained for larvae reared at 16˚C.

As expected, the food treatments did not affect the first week's survival, when all larvae were fed similarly and larvae still rely on their yolk reserves. Thus, the overall survival among treatments was very similar during the first two weeks (Fig 3). Larvae reared at 20˚C had a period of acclimatization that started at day 6 of the experiment (larvae aged 8 dph) when the temperature was progressively increased by 1 degree per day, ending on day 10 (12 dph). The similar trends in survival among treatments obtained during the first week of the experiment are most likely a result of the acclimation period. However, from the second week onwards (from day 8 to 15 in Fig 3), there is an important decrease in survival, with both temperature treatments following the same trend, *i.e.*, a sharp decrease in the survival rate for all treatments that stabilizes around day 18 (at 20 dph, Fig 3). The lowest percentage for total survival was obtained for LowFood treatment at 20˚C (Table 2). For comparing the survival rates and their relationship to food availability, we fitted linear regressions for the survival curves during the period where the decrease in survival was the highest (Fig 3b). The results show that survival was significantly different between temperature treatments in the LowFood treatment (no overlapping on the 95% confidence intervals).

For the whole experiment duration, a Mantel-Cox test on survival curves indicates no significant differences among feeding frequencies or between rearing temperatures (Mantel-Cox, p-value > 0.05, see S1 Fig in S1 File).

**Table 2. Lethal time for 50% ($LT_{50}$, days)** *Solea solea* **larvae reared at two temperatures with different feeding frequencies and total survival at the end of the experiment.**

| LT50 (days) | | | |
|---|---|---|---|
| Temp (˚C) | HighFood | MediumFood | LowFood |
| 16 | 12.4 | 12.1 | 12.2 |
| 20 | 10.2 | 9.5 | 10.6 |
| Total survival at the end of experiment (%) | | | |
| 16 | 28 | 42 | 39 |
| 20 | 39 | 44 | 25 |

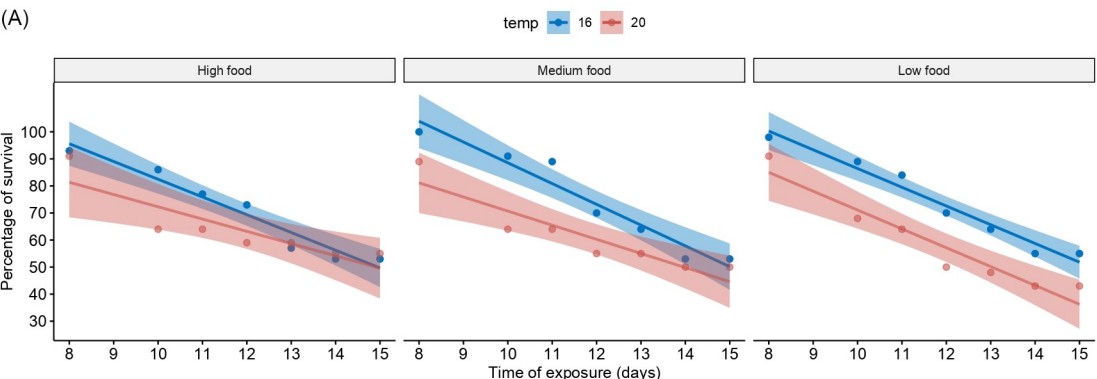

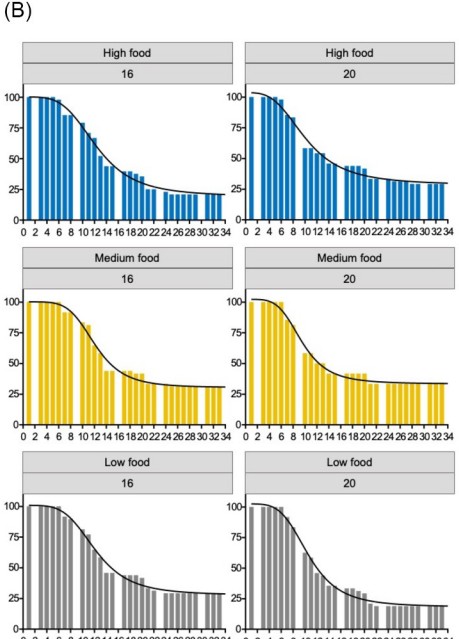

**Fig 3. Larvae survival during the experiment.** Percentage of survival with fitted four parameter logistic regressions in *S. solea* according to the feeding frequency at two different temperatures. a: whole experiment duration (33 days larvae aged 35 dph). b: Solid lines represent linear models fitted to the to the second's week percentage of survival with 95% confidence level intervals (shadow area).

## Growth

The interaction between temperature and feeding frequencies does not explain the observed variation in the total length of larvae (Fig 4, S3 Table in S1 File). However, the feeding frequency factor did affect larval length independently (p-value$_{feeding}$ < 0.001) while temperature did not have a significant effect on growth for, at least, the first 35 days following hatching.

For the effect of feeding frequencies, a pairwise test showed that larvae from the HighFood treatment were significantly longer than larvae fed two times per week (LowFood p-value < 0.01). No significant differences were observed between HighFood and MediumFood or MediumFood and LowFood treatments.

By the end of the experiment, the longest larvae were from the HighFood treatment reared at 20˚C, accounting for an average length of 9.8 ± 0.6 mm (Table 3).

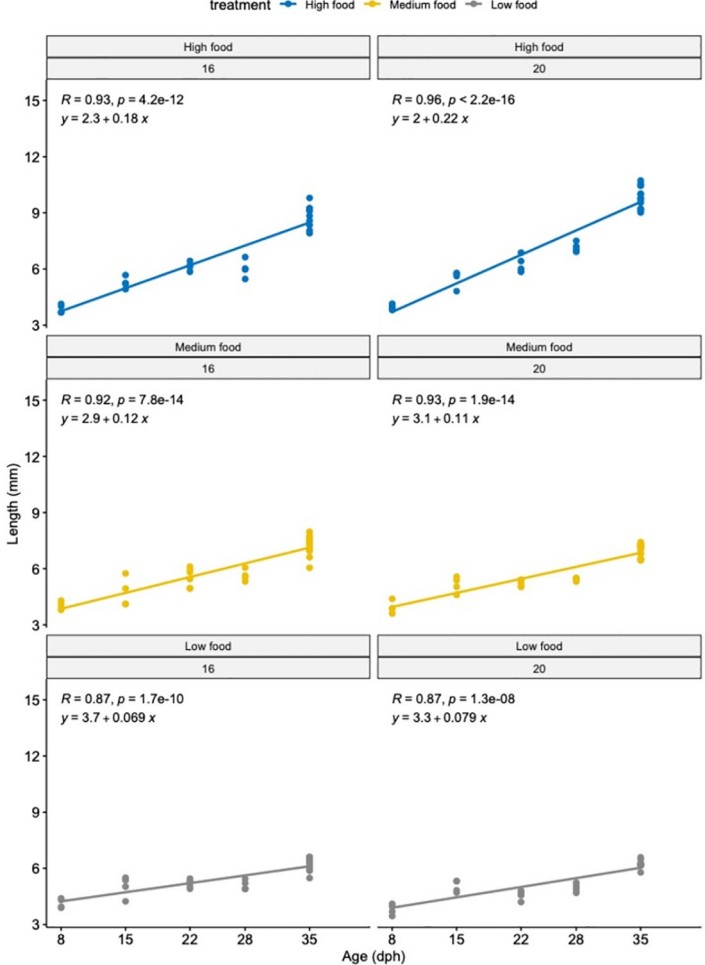

**Fig 4. Length of sole (mm) larvae reared at different temperatures and different feeding frequencies in experiment 2.** Left panels: larvae reared at 16°C. Right panels: larvae reared at 20°C at high (blue), medium (yellow) and low (grey) feeding frequencies.

Given the high mortality we observed during the first 3 weeks (Fig 3), we decided to stop sampling larvae for quantifying the dry weight of larvae. Thus, there are no dry weight records during the fourth, and fifth weeks of the experiment.

Regarding dry weight data, none of the tested factors or their interaction explained the observed variation (Fig 5, S3 Table in S1 File).

In the HighFood treatment alone, we observed the same pattern regardless of the rearing temperature, showing a decrease in dry weight from day 8 to 15, followed by a steep increase

**Table 3. Summary of biometric data at the end of the experiment (35 dph).**

| Length (mm) | | | |
|---|---|---|---|
| Temp (°C) | HighFood | MediumFood | LowFood |
| 16 | 8.8 ± 0.6 | 7.5 ± 0.2 | 6.2 ± 0.9 |
| 20 | 9.8 ± 0.6 | 7 ± 0.3 | 6.2 ± 0.5 |

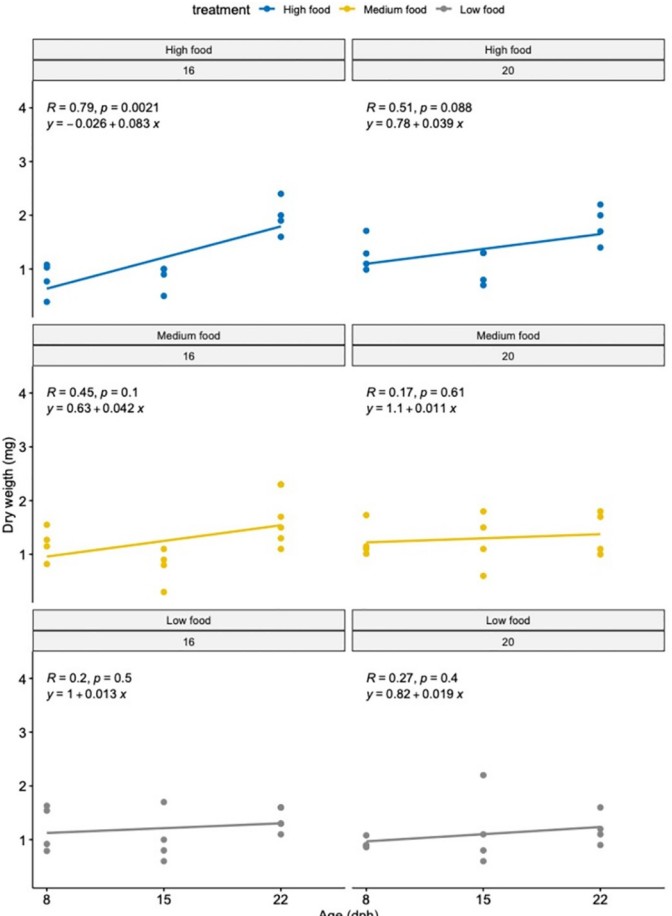

**Fig 5. Dry weight of sole (mm) larvae reared at different temperatures and different feeding frequencies in experiment 2.** Left panels: larvae reared at 16˚C. Right panels: larvae reared at 20˚C at high (blue), medium (yellow) and low (grey) feeding frequencies.

in weight towards day 22 and 35. For the other two feeding frequencies, data showed little variation from the start until the end of the experiment (Fig 5).

## Metamorphosis

The onset of metamorphosis, did change between treatments, occurring at a younger age for larvae reared at 20˚C than larvae reared at 16˚C (Fig 6). The first larva that started metamorphosis, was an individual from the HighFood treatment reared at 20˚C, starting the process at 19 dph. At 16˚C, the first larva that started metamorphosis was at 22 dph.

Only individuals from HighFood finished metamorphosis. MediumFood larvae, began their metamorphosis at 33 dph and 35 dph at 20˚C and 16˚C, respectively. LowFood larvae reared at 20˚C, started the metamorphosis around 35 dph, and those reared at 16˚C did not advanced their metamorphosis beyond stage 0 (Fig 6).

A total of 5% of the surviving larvae showed abnormal metamorphosis, often involving the non-migration of the left eye (S2 Fig in S1 File). All metamorphosis abnormalities were obtained in the HighFood treatment at 20˚C, which was also the treatment with the highest proportion of individuals metamorphosing.

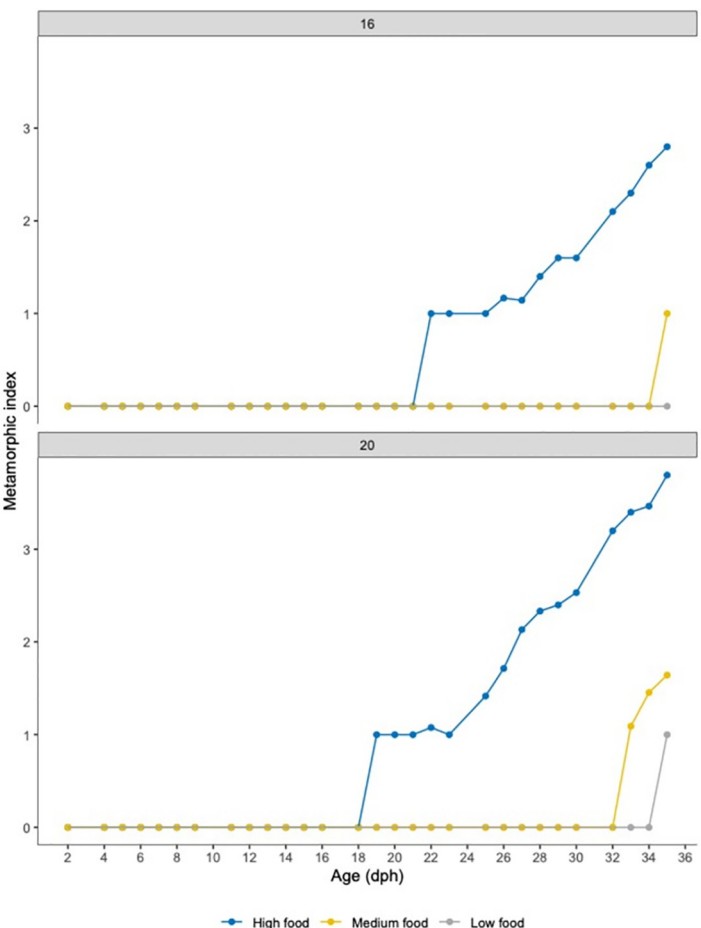

**Fig 6. Metamorphic index in *Solea solea* larvae.** Differences in the metamorphosis onset and development of larvae reared at two temperatures and 3 different feeding frequencies (high, medium and low food).

By the end of the experiment in HighFood, the average size of larvae at stage 4 was higher (but not significant) for larvae reared at 20˚C (10.9 mm ± 0.4, n = 8) vs. those reared at 16˚C (9.4 mm ± 0.3, n = 3). Further, the first larva that finished metamorphosis were 10.1 mm long, and 9.4 mm for 20˚C and 16˚C, respectively. Larvae in the MediumFood treatment did not complete metamorphosis and the average size of stage 2 larvae when reared at 20˚C was slightly smaller (7.11 mm ± 0.3, n = 9) vs. those larvae reared at 16˚C (7.54 ± 0.2, n = 8), which were still on stage 1.

## Discussion

### Methodological insights

In aquaculture and ecotoxicological research, the classical set-up for rearing/exposing fishes on their early life stages involves larvae being group-housed in tanks or cylinders. In 2017, De Swaef and collaborators published a protocol where sole larvae were reared using 24-well microplates as a housing system. Their efforts in sole larviculture were primarily oriented toward improving larvae survival while developing a sterile gnotobiotic environment. Their results highlighted that working within a system that allows rearing larvae individually is a

great advantage. Using microplates as a housing system guarantees better water quality unin-fluenced by dead individuals. Further, it allows monitoring each individual's growth, behavior, and condition, while contributing to both "Reduction" and "Refinement" objectives within animal ethics.

In this work, we simplified the proposed protocol from De Swaef et al. (2017) by removing water and eggs disinfection steps and sterilizing food (*Artemia salina*) and materials, which we considered unnecessary for reaching the objectives of our work. Further, we did not use a microbiological safety cabinet, and although we worked with filtered artificial seawater, the water was not autoclaved.

Although intuitive thinking might indicate that reducing the sterile conditions of the proto-col would impact survival rates, the differences in survival between De Swaef controls and ours are not that high. The reported survival percentage in four different batches (spawning batches) at 26 dph were 85%, 84%, 65%, and 58% [20]. Another work from the same authors reported a survival percentage at 17 dph of 90% [34]. During our pilot, we obtained a similar total percentage of survival of 88% and 82% at 17 and 26 dph, respectively (S3 Fig in S1 File). On the contrary, the survival reported during our second pilot and the experiment in the con-trol condition (*i.e.*, HighFood at 16˚C) is much lower, with a total survival of 48% at 17 dph.

The accentuated differences we obtained in the survival of control larvae between pilot and experiment could be explained by the quality of the eggs. The same observation applies for sur-vival during the second pilot compared to the experiment (S4 Fig in S1 File). Indeed, our observation during the second pilot -later confirmed with data- was that very few larvae started feeding, indicating that the transition between endogenous to exogenous feeding was not suc-cessful for most larvae, regardless of the treatments. In terms of reduced survival, following a 5 days delayed first feeding, a work by Lagardère et Chaumillon [35] also obtained very low sur-vival rate (1.3%) after 22 days of experiment. At 33 dph, total survival in the pilot experiment was 81%, whereas, during the experiment, all treatments and temperatures considered were below 50% at the same age.

Temperature changes affect physiology, gamete development, and maturation, typically ending in poor sperm and oocyte quality, albeit gamete quality is crucial for developmental success in the next generation [36]. The weather conditions in the Netherlands in 2021 were not favorable for suitable egg-laying. First, the first spawning occurred late, followed by a cold wave that disrupted and stopped the laying for several days. The eggs obtained for the second pilot came from one of the first spawning events after the weather disruption, whereas the eggs from the experiment were collected after a couple of weeks of stable weather. The differences in quality and fertilization were evident. On the contrary, weather conditions in 2019 were favorable for egg-laying, supporting the hypothesis that egg quality contributed to the observed differences between the results from the experiment in 2019 and those from 2021.

The critical decrease in survival occurring during the second week corresponds to the tran-sition from endogenous to exogenous feeding. The abundance of food plays an essential role in determining the survival of first-time feeding larvae. This is because, after yolk exhaustion, fish larvae must establish themselves as active feeders, or they risk starvation [37]. Generally, the time span from yolk exhaustion to starvation is temperature-dependent and species-spe-cific. Effects of feeding frequency in larvae survival have already been studied in several species including *Oreochromis niloticus*, *Lates calcarifer*, and *Heterobranchus longifilis* [38]. Atsé and collaborators (2012) tested the effects of feeding rates (25%, 50%, 75%, and 100% of biomass) and feeding frequencies (one meal per day either in the morning or in the afternoon, two meals per day, and three meals per day) on survival and growth of *Heterobranchus longifilis* larvae over a rearing time of 28 days. They showed that growth and survival were proportional to increasing feeding rate. Also, survival and growth were highest in larvae fed three times a

day, and cannibalism was less important with increasing feeding rates. All these results allowed them to conclude that optimal conditions for rearing this species include a feed ration of 100% and a feeding frequency of three meals per day.

In the case of sole metamorphic larvae reared under laboratory conditions, density, ration, or feeding frequency and the timing for the first feeding can influence the survival and growth of larval fish. Further, the amount of food required for the development of fish under laboratory conditions varies mainly according to species, size, and rearing conditions -notably temperature- and the ideal feeding rate and frequency generally decrease as the fish grows [39].

Low feeding frequency or starvation and increased temperature can increase mortality. As suggested for the Japanese flounder, the relatively higher mortality at 20°C could be explained because the metabolic and energy costs are higher than that of sole reared at 16°C. As a result, feeding becomes more difficult as the larvae have to spend energy, which they have little, to hunt, particularly in those treatments where food availability is lower [37].

The protocol adjustments made within this work make it a much more accessible, reproducible and affordable protocol, suitable for exploring the single effects of feeding regimes and temperature and the combined effects of temperature and future chemical exposure.

Finally, the assumption of high survival in control conditions is pure logic from an experimental point of view. However, from an evolutionary perspective, the early-life survival of fish larvae is expected to be low. As an r-strategist, the recently hatched *S. solea* larvae (and most fish larvae, with few exceptions like viviparous sharks) are very often expected to die, either by predation, changes in environmental cues, starvation, or simply because the quality of the egg was initially low. Thus, in our opinion, it is maybe too ambitious to expect survival rates higher than 60% (threshold proposed in the OECD guidelines) in this species, and it is necessary to interpret the experimental results considering factors such as initial egg quality.

## Effects of temperature and feeding on life history traits

The hypothesis that lower food availability will delay the start of metamorphosis while temperature increase will fasted up growth was not refuted. However, our data also reflects the complexity of interpreting these results as the effects of these factors are highly interconnected.

In our experiment, the largest larvae and the faster growth rates were those from the HighFood treatment reared at 20°C, and the start of metamorphosis was affected by feeding frequency, with HighFood larvae starting it 14 and 16 days before MediumFood and LowFood treatments, respectively.

A current review on the effects of temperature and contamination on *S. solea* larvae ontogenesis evidenced that only a handful of studies have focused on understanding the effects of temperature on spawning and larvae development (See section 3.1 in Sardi et al., 2021). However, the before-mentioned review focuses on works from 2003, thus neglecting fundamental studies made much earlier often under the umbrella of aquaculture research. These include the works from Baynes and Howell in 1996 and Fonds in 1971 [40, 41]. The last studied, under laboratory conditions, the impact of temperature on larvae ontogenesis. Results showed that all larvae started metamorphosis at length between 6–7 mm regardless of the temperature they were reared at (13, 16, and 19°C). Therefore, since temperature promotes growth, metamorphosis at 19°C started earlier than at 16°C, which in turn was earlier than at 13°C [41]. Our results are in agreement with the previous results. However, by the end of metamorphosis, larvae from the 19°C treatment were 1 mm shorter than those reared at lower temperatures [41].

Life-history theory suggests that the interactions and trade-offs between traits related to survival, growth, and reproduction describe the speed of life. As such, a fast life-history species is characterized by faster growth, higher reproductive output, shorter gestation times, earlier

ages at maturity, shorter lifespans, shorter generation times, smaller body sizes, and higher population growth rates, while a slow life history species lies in the opposite, constituting the life-history continuum [42].

Research from Wang and collaborators (2020) evidenced that sea surface temperature increase accelerates life-history traits in over 332 indo-pacific fish species, highlighting that changes at the population level will depend on the life-history continuum, which varies among species. According to their results, water temperature generally shifted species towards faster life histories, causing faster growth rates, higher natural mortality rates, smaller asymptotic lengths, and earlier maturation [43]. Our results for the common sole sustain the last observations about faster growth rates, faster life histories -notably regarding metamorphosis, which is advanced in larvae reared at 20°C- and higher natural mortality rates when food density is low.

At higher temperatures, the metabolic rate of larvae is expected to increase, which means that fish have higher basal energy demands and larval growth rates increase. Although responses vary considerably among species, mass-specific oxygen consumption in fish larvae —a proxy of metabolic rate- has been proven to increase with temperature [44]. Regarding larval growth rates, empirical observations show that growth rates tend to be approximately linear until the lethal upper thermal limit is reached, at least in most species studied to date [45]. The previous statement contrasts with thermal reaction norms -continuous functions describing the relationship between an environmental variable (e.g., temperature) and the phenotype expressed by a given genotype [46]- of growth in juveniles and adults, which usually decline well before the lethal thermal limit is reached. The temperature range for the successful development of *S. solea* eggs is between 7 to 19°C, 10 to 23°C for larvae, and 7 to 27°C for juvenile growth [41, 47]. Thus, under the worst RCP scenario temperature predictions, we might expect that larval growth rates will tend to be maintained, whereas effects on juvenile growth -such as shorter asymptotic lengths- might be observed. Our results support the last, as the slopes for size and weight measurements in both the HighFood and MediumFood treatments were similar for larvae reared at optimal and +4°C (Fig 4).

Changes in the pelagic larval duration (PLD) and potential mismatch between hatching and the zooplankton cycle timing are far more preoccupying effects of temperature increase than changes in larvae and juvenile growth rates. Indeed, the times until yolk absorption, metamorphosis and PLD of fish larvae are all negatively correlated with temperature [45], observations also evidenced by our results. Because mortality is generally very high during the larval phase, faster growth and shorter PLD at higher temperatures could positively affect larval survivorship [45]. In our experiments, the survival of larvae reared at 20°C was higher than that of larvae at 16°C for the HighFood treatment, supporting the previous assumption.

In this work we did not study the effect of temperature on spawning. Our results evidenced that larvae reared at higher temperatures have a shorter PLD, suggesting that higher temperatures due to climate change might favor larvae survivorship. However, during our pilot experiment we observed massive mortality in all treatments, which could also be related to larvae missing their first feeding due to low prey densities or reduced capabilities in capturing prey (linked to egg quality), making them more susceptible to predation and other sources of mortality [48]. The last result supports the critical period hypothesis in which first-feeding survival larvae define year-class strength and fish recruitment success.

Food availability is compulsory for sustaining higher development and growth in warmer waters. As climate models predict an increase in water temperature and lower food availability, the potential increase in survival probability of fish larvae due to faster growth and shorter PLD that reduces the predation pressure (as proposed in the size/growth hypothesis) would be counterbalanced by the increase in mortality during the critical period due to lower prey densities.

Given that changes in phenology vary among functional groups—as evidenced by [49]-warming has led to mismatch between trophic levels, while changing the synchrony of timing between primary, secondary, and tertiary production. However, understanding if a phenological modification is a direct response to environmental modifications (temperature advancing spawning), an indirect response to prey availability (climate change impacting primary and secondary production), or a combination of both remains a challenge.

### Metamorphosis success

Metamorphosis is a stage that implies significant metabolic changes, including the slowing down of metabolic activity, massive morphology remodeling and the re-calibration of vision to detect prey. Altogether metamorphosis is a stage that requires a lot of energy. If food densities are low, growth and metamorphosis takes longer, which increases the vulnerability of the larvae to starvation and predation [50, 51]. Further, metamorphosis implies the transition to new habitats, exposing the young fish to new forms of competition and predation. Also, the colonization of young sole into unstable nursery ecosystems situated in bays and estuaries may be crucial to the survival of the new settlers due to the variable hydrological conditions in the nurseries, predation, and competition for space and resources. This context has pushed many authors to believe that the year-class strength of several fish species is generally determined either during or just after metamorphosis [52].

In this work, we experimentally tested the effects of food and temperature -two important environmental drivers- in sole metamorphosis success separately, but *in natura*, it is unlikely that they operate independently. As previously discussed, temperature-driven changes in phenology could cause a mismatch between different trophic levels and negatively affect recruitment not only by directly increasing mortality but also by extending the metamorphic period. Because the onset of metamorphosis requires larvae to acquire a competent size, when conditions are not favorable for growth, metamorphosis takes longer and renders larvae vulnerable to predation. On the contrary, a higher temperature without food limitation could be an advantage as it would speed up the metamorphosis process. However, there is still the risk of finding fully metamorphosed juveniles outside nursery areas, and as such, still vulnerable to predation and in an environment low in preys.

Although the interaction between feeding and temperature was not statistically significant in both experiments, it was the LowFood treatment at 20°C, the treatment with the highest mortality. Larvae growing at higher temperatures have higher metabolic rates and higher energetic demands, but they will die if food is unavailable to meet these demands.

### Conclusions

The objective of this study was to determine the impact of different feeding frequency regimes -as a proxy for food availability- on common sole larvae development in the context of global warming. For this, we hypothesize that 1) feeding frequencies would impact the survival, growth, and metamorphosis success of the larvae and 2) rearing temperature would increase the observed effects by adding additional physiological stress to larvae reared at +4°C.

Accounting for our results, we cannot entirely accept the first hypothesis, as we did not find a statistically significant relationship between survival and the tested treatments. However, we observed significant effects of the feeding frequency factor on larvae growth (measured as length and dry weight), and there were differences in the start and pace of metamorphosis. The more we fed the larvae, the longer and heavier they were. Regarding the metamorphosis of the larvae, we could see that there was a delay in the onset of metamorphosis. Indeed, the larvae reared at 16°C and fed HighFood begun their metamorphosis ten days before Medium

Food larvae and 17 days before LowFood larvae. At 20˚C, the start of metamorphosis was even faster, with HighFood larvae beginning their metamorphosis 14 days before MediumFood ones, and 16 days before LowFood larvae.

Overall, the feeding frequency regimes impact growth and metamorphosis but not survival. However, this is only true if the larvae are fed *ad libitum* during the first week. During our second pilot, the larvae were not fed until satiation, altering the transition from endogenous to exogenous feeding and leading most larvae to die. Based on the experiment results, our hypothesis of feeding frequency regimes altering growth and metamorphosis should be accepted.

Summarizing, food availability impacts the growth and onset of metamorphosis in *S. solea* larvae, while higher temperature advanced the onset of metamorphosis and increased the occurrence of abnormalities.

## Future perspective

Experimentally testing the effects of any stressing factor is always a challenge, mainly because rearing and keeping animals in laboratory conditions is already very stressful for most of them. Here we aimed at testing a user-friendly protocol for evaluating the effects of food availability and temperature on the larval development of the common sole. We focused in larval phase, as there is a substantial research library on sole juveniles, with less information linking larvae survival and juvenile recruitment. Despite conditions being far from perfect, we propose a reproducible experiment that provides insights into the health and survival of larvae exposed to conditions mimicking future climate change scenarios. Recommended modifications for the proposed protocol include 1) using 12-well microplates for providing a larger volume and area to larvae, and 2) providing enriched food with higher nutritious value, which would allow identifying caveats of the experiment related to energy while minimizing sources of bias such as fish behavior.

Climate change impacts might not operate independently from toxicity effects from pollutants. We consider that the protocol proposed here is suitable for replacing standard ecotoxicological testing used for aquatic ecological risk assessment, such as the fish early-life stage (FELS) test [53]. The last is the most frequently used bioassay for fish toxicity, supporting aquatic ecological risk assessments and chemical management programs. However, this test guideline requires an average post-hatch control survival of at least 75% in animals usually housed in the same chamber. These conditions are fulfilled for some model fish species, which are often very robust laboratory species that might have already undergone adaptations to laboratory conditions (following multiple generation rearing), such as rainbow trout, zebrafish, Japanese medaka, and fathead minnow. However, when the interest species is not listed in the recommended annex from the OECD guidelines, the chances are that survival higher than 60% in control will not be met. Thus, chemical testing in early life stages on fish is strongly biased towards these model species, which are all freshwater species, and much less is known about the impact of chemicals in marine species like the common sole (see section 3 in 10 for a summary of studies conducted in *S. solea* with an ecotoxicological focus).

There is a need for testing chemical toxicity in other less robust species, and one way to reach it is by developing or adapting current methods for assuring higher survival rates. Individual housing in microplate wells increased larvae survival, which is an appropriate experimental improvement. Standardizing a protocol that allows evaluating the effect of contaminants in this species is relevant given its economic and ecological importance and its susceptibility to chemical and fishing pressures.

## Supporting information

**S1 File.**
(DOCX)

**S1 Data.**
(XLSX)

## Acknowledgments

The authors wish to thank Dr. Véronique Loizeau, Dr. Pierre Labadie and Dr. Florence Mounier for the fruitful conversations we held, which helped with the conceptualization of this work.

## Author Contributions

**Conceptualization:** Adriana E. Sardi, Marie-Laure Bégout, Xavier Cousin, Hélène Budzinski.

**Data curation:** Adriana E. Sardi, Anne-Laure Lalles, Xavier Cousin.

**Formal analysis:** Adriana E. Sardi, Anne-Laure Lalles, Xavier Cousin.

**Funding acquisition:** Adriana E. Sardi, Hélène Budzinski.

**Methodology:** Adriana E. Sardi, Marie-Laure Bégout, Xavier Cousin.

**Project administration:** Hélène Budzinski.

**Supervision:** Marie-Laure Bégout, Xavier Cousin, Hélène Budzinski.

**Writing – original draft:** Adriana E. Sardi.

**Writing – review & editing:** Marie-Laure Bégout, Xavier Cousin.

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
