## [Decision Letter · Decision Letter 0]

14 Dec 2022

PONE-D-22-19553Temperature and feeding frequency impact the survival, growth, and metamorphosis success of Solea solea larvaePLOS ONE

Dear Dr. Sardi

Thank you for submitting your manuscript to PLOS ONE. After careful consideration, we feel that it has merit but does not fully meet PLOS ONE’s publication criteria as it currently stands. Therefore, we invite you to submit a revised version of the manuscript that addresses the points raised during the review process.

The two assessments suggested commonalities and should be accepted or addressed in a separate letter. I believe that the modifications are quite simple and can be performed quickly.

We look forward to receiving your revised manuscript.

Kind regards,

Sergio N. Stampar, Dr.

Academic Editor

PLOS ONE

Journal Requirements:

2. In your Methods section, please provide additional information on the animal research and ensure you have included details on : (1) methods of sacrifice (2) methods of anesthesia and/or analgesia, and (3) efforts to alleviate suffering.

“This study was funded by the IdEx Bordeaux International Post-doctorates program attributed to A. E. Sardi. The authors wish to thank Dr. Véronique Loizeau, Dr. Pierre Labadie and Dr. Florence Mounier for the fruitful conversations we held, which helped with the conceptualization of this work. “

“A.S. was funded by the IdEx Bordeaux International Post-doctorates program. The funders had no role in study design, data collection and analysis, decision to publish, or preparation of the manuscript”

9 We noticed you have some minor occurrence of overlapping text with the following previous publication, which needs to be addressed:

- https://www.sciencedirect.com/science/article/abs/pii/S0304380020302945?via%3Dihub

In your revision ensure you cite all your sources (including your own works), and quote or rephrase any duplicated text outside the methods section. Further consideration is dependent on these concerns being addressed.

Additional Editor Comments (if provided):

Dear Dr. Sardi

We just received two reviews of your manuscript and both suggested minor modifications. Please address these changes or explain why you are not making changes in a separate letter.

After this step, we will be able to proceed with the evaluation of your manuscript.

Kind regards

Sergio Stampar

Reviewers' comments:

Reviewer's Responses to Questions

**Comments to the Author**

1. Is the manuscript technically sound, and do the data support the conclusions?

Reviewer #1: Yes

Reviewer #2: Yes

2. Has the statistical analysis been performed appropriately and rigorously? 

Reviewer #1: Yes

Reviewer #2: Yes

3. Have the authors made all data underlying the findings in their manuscript fully available?

Reviewer #1: Yes

Reviewer #2: Yes

4. Is the manuscript presented in an intelligible fashion and written in standard English?

Reviewer #1: Yes

Reviewer #2: Yes

5. Review Comments to the Author

Reviewer #1: The manuscript “Temperature and feeding frequency impact the survival, growth, and metamorphosis success of Solea solea larvae” by Sardi et al is a laboratory experiment with S. solea larvae targeting to endpoints such as survival, growth and metamorphosis.

The manuscript is well written and carefully justified in their procedure and findings. The authors also use a novel approach (proposed by other authors using 24 well plates) to be applied in ecotoxicology of larvae. They also propose a marine species as sentinel, other than the most frequently used fresh water model fish. The use of a more realistic natural species with high economic interest validates their effort as it could be applied in aquaculture and more particularly in larviculture. Nevertheless, their study also has some limitations such as the low quality of the stock egg production in some of their trials. They justify this as naturally occurring phenomena also in natural populations.

Their findings are not conclusive but their protocol and justification seem sound to me. Overall it is evident the large number of variables to be considered in laboratory experiments trying to predict biological consequences due to environmental variables fluctuations.

Other minor suggestions are:

Line 298. The sentence seems unfinished.

Fig 2. The results for Temp of 20ºC are missing.

Fig 3 and 4 might be better replaced by a table informing on the characteristics of the slopes, correlations and significances under the different temperatures and feeding regimes in a more comparative way that the Figures reflect.

Reviewer #2: Overall, this study was well designed and well written. I think it will contribute to this field.

However, some parts need minor revisions.

I will summarize some of them below,

The expressions in the abstract part but not suitable for this part should either be rearranged or moved to the relevant place.

There are some errors in the use and spelling of references in some parts. These need updating and editing.

Adding photos of Solea solea larvae as figures can support the study visually.

Other recommendations are detailed in the attached file.

6. PLOS authors have the option to publish the peer review history of their article (what does this mean?). If published, this will include your full peer review and any attached files.

Reviewer #1: No

Reviewer #2: No

---

## [Author Response · Author response to Decision Letter 0]

19 Dec 2022

Dear Dr. Stampar, 

Please find below the reply to all the reviewers’ comments specifying the modifications made in the revised manuscript as also the modifications made to meet Plos One format style. 

Reviewer #1: The manuscript “Temperature and feeding frequency impact the survival, growth, and metamorphosis success of Solea solea larvae” by Sardi et al is a laboratory experiment with S. solea larvae targeting to endpoints such as survival, growth and metamorphosis.

The manuscript is well written and carefully justified in their procedure and findings. The authors also use a novel approach (proposed by other authors using 24 well plates) to be applied in ecotoxicology of larvae. They also propose a marine species as sentinel, other than the most frequently used fresh water model fish. The use of a more realistic natural species with high economic interest validates their effort as it could be applied in aquaculture and more particularly in larviculture. Nevertheless, their study also has some limitations such as the low quality of the stock egg production in some of their trials. They justify this as naturally occurring phenomena also in natural populations.

Their findings are not conclusive but their protocol and justification seem sound to me. Overall it is evident the large number of variables to be considered in laboratory experiments trying to predict biological consequences due to environmental variables fluctuations.

Response: Thank you for your comments and your positive opinion on our work, which as you has mentioned targets to validate laboratory experiments for ecotoxicology, larviculture and in the general to contributing on using marine fish species as model species. 

Other minor suggestions are:

Line 298. The sentence seems unfinished.

Response: Indeed, it was incomplete, thank you for noticing.

Fig 2. The results for Temp of 20ºC are missing.

Response: Apologies for this mistake, it seems to be an error while uploading the figure. 

Fig 3 and 4 might be better replaced by a table informing on the characteristics of the slopes, correlations and significances under the different temperatures and feeding regimes in a more comparative way that the Figures reflect.

Response: We appreciate the reviewer suggestion, but consider that for our objective —comparing the growth and weight among treatments— a figure is more appropriate than a table. Using figures allows for rapidly visualizing the differences, particularly because the axis limits are the same for all the figures, which gives a common ground for easily comparing the slopes. 

Reviewer #2: Overall, this study was well designed and well written. I think it will contribute to this field.

Response: Thank you for your time and for supporting our research.

However, some parts need minor revisions.

I will summarize some of them below,

The expressions in the abstract part but not suitable for this part should either be rearranged or moved to the relevant place.

Response: We have placed the highlighted sentence in the first paragraph of the introduction as suggested. 

There are some errors in the use and spelling of references in some parts. These need updating and editing.

Response: Updated, thank you for noticing. 

Adding photos of Solea solea larvae as figures can support the study visually.

Other recommendations are detailed in the attached file.

Response: We have included a figure of a fully metamorphosed larva in the revised version. 

Other formatting request from Plos One:

In addition to the comments from the reviewers, we have done the following formatting modifications in line with PlosOne guidelines. 

1) Change the size of headings and subheadings to 18 and 16 pt respectively

2) Remove all points after the Fig citations

3) Bold the figure number and title of all figure legends. Include a figure title. 

4) Move the figure legends immediately after the first paragraph citating the figure 

5) Remove the figures from the manuscript file 

6) Moved the tables from the end to immediately after the first citation

7) Swapped the order when citing figures or tables from the supplementary information document (i.e., S1 Fig instead of Fig S1)

2. In your Methods section, please provide additional information on the animal research and ensure you have included details on : (1) methods of sacrifice (2) methods of anesthesia and/or analgesia, and (3) efforts to alleviate suffering.

We have included details about the methods of sacrifice and concentration and chemical for anesthesia (line 239). 

This is not our case.

We have included the necessary data on supplementary information as an xls file. 

Done

“This study was funded by the IdEx Bordeaux International Post-doctorates program attributed to A. E. Sardi. The authors wish to thank Dr. Véronique Loizeau, Dr. Pierre Labadie and Dr. Florence Mounier for the fruitful conversations we held, which helped with the conceptualization of this work. “

“A.S. was funded by the IdEx Bordeaux International Post-doctorates program. The funders had no role in study design, data collection and analysis, decision to publish, or preparation of the manuscript”

The funding statement is correct. I have removed the funding information from the acknowledgements section. 

The Research and High Education French Ministry evaluated and approved all animal experimentations. The committee from the department of practical research activities licensed the project “Flatfish adaptation to temperature, feeding and pollution stress” led by Dr. Marie-Laure Bégout from Ifremer under the number reference APAFIS #38190-202208082103879. 

This section is now included in the revised version of the manuscript. 

The list of captions for the supporting tables and figures are now included at the end of the revised manuscript.

9 We noticed you have some minor occurrence of overlapping text with the following previous publication, which needs to be addressed:

- https://www.sciencedirect.com/science/article/abs/pii/S0304380020302945?via%3Dihub

In your revision ensure you cite all your sources (including your own works), and quote or rephrase any duplicated text outside the methods section. Further consideration is dependent on these concerns being addressed.

Done

---

## [Editor Report · Decision Letter 1]

18 Jan 2023

Temperature and feeding frequency impact the survival, growth, and metamorphosis success ofSolea solealarvae

PONE-D-22-19553R1

Dear Dr. Sardi,

We’re pleased to inform you that your manuscript has been judged scientifically suitable for publication and will be formally accepted for publication once it meets all outstanding technical requirements.

Kind regards,

Sergio N. Stampar, Dr.

Academic Editor

PLOS ONE

Additional Editor Comments (optional):

Thank you again for your interest in publishing your manuscript in PLOS One.

After a detailed review of the modifications made by the authors in response to the suggestions, I am pleased to inform you that the manuscript can be accepted for publication.
---

## [Editor Report · Acceptance letter]

20 Jan 2023

PONE-D-22-19553R1 

Temperature and feeding frequency impact the survival, growth, and metamorphosis success of *Solea solea* larvae 

Dear Dr. Sardi:

I'm pleased to inform you that your manuscript has been deemed suitable for publication in PLOS ONE. Congratulations! Your manuscript is now with our production department. 

Kind regards, 

on behalf of

Dr. Sergio N. Stampar 

Academic Editor

PLOS ONE